# ETMT: A Tool for Eye-Tracking-Based Trail-Making Test to Detect Cognitive Impairment

**DOI:** 10.3390/s23156848

**Published:** 2023-08-01

**Authors:** Jyotsna Chandrasekharan, Amudha Joseph, Amritanshu Ram, Giandomenico Nollo

**Affiliations:** 1Department of Computer Science and Engineering, Amrita School of Computing, Amrita Vishwa Vidyapeetham, Bengaluru 560035, India; j_amudha@blr.amrita.edu; 2Department of Industrial Engineering, University of Trento, 38123 Trento, Italy; giandomenico.nollo@unitn.it; 3HCG Cancer Center, Bangalore 560002, India; amrit.r@hcgel.com

**Keywords:** eye tracking, trail-making test, adaptive neuro-fuzzy-inference system, cognitive impairment, k-means clustering, fuzzy-inference system, inattentional blindness

## Abstract

The growing number of people with cognitive impairment will significantly increase healthcare demand. Screening tools are crucial for detecting cognitive impairment due to a shortage of mental health experts aiming to improve the quality of life for those living with this condition. Eye tracking is a powerful tool that can provide deeper insights into human behavior and inner cognitive processes. The proposed Eye-Tracking-Based Trail-Making Test, ETMT, is a screening tool for monitoring a person’s cognitive function. The proposed system utilizes a fuzzy-inference system as an integral part of its framework to calculate comprehensive scores assessing visual search speed and focused attention. By employing an adaptive neuro-fuzzy-inference system, the tool provides an overall cognitive-impairment score, allowing psychologists to assess and quantify the extent of cognitive decline or impairment in their patients. The ETMT model offers a comprehensive understanding of cognitive abilities and identifies potential deficits in various domains. The results indicate that the ETMT model is a potential tool for evaluating cognitive impairment and can capture significant changes in eye movement behavior associated with cognitive impairment. It provides a convenient and affordable diagnosis, prioritizing healthcare resources for severe conditions while enhancing feedback to practitioners.

## 1. Introduction

People’s lives have changed significantly in modern times, and some may find it challenging to recall details, understand novel information, remember, pay attention to details, or make judgments that may affect their daily lives [1]. The subtle changes in human beings’ cognitive functions influence a person’s behavior. If a person has trouble remembering, learning new things, and concentrating on their work, their cognitive ability is deteriorating or affected by cognitive impairment [2]. Cognitive impairment is regarded as one of the most costly diseases, considering the cost of drugs and nursing facilities [3]. Cognitive impairment is considered an incurable disease [4,5]. However, the growth of the disease can be decreased by providing adequate treatment and care if it can be diagnosed in an earlier stage.

Nowadays, there is a considerable increase in people suffering from this disability [6], and the rapid increase in people with dementia has become a significant health issue. However, the progression of cognitive impairment can be slowed down with early detection and prompt treatments [7]. Though it is mainly observed after the age of 65, it is not limited to any specific age group. Other risk factors for cognitive impairment include family history, injury to the brain, exposure to toxicants, brain irradiation [8], education level, and other diseases. The side effects of some medications, deficiency of vitamins, depression, and other health issues can also be the reason for mild cognitive impairment (MCI) [9].

Cognitive impairment ranges from mild to intense. The transitional stage from subtle cognitive abnormalities to the early stages of dementia is known as MCI [10,11]. People with mild disabilities have minute changes in their cognitive function but can manage their daily activities. Intense levels of impairment can result in a loss of the ability to speak, write, and understand the meaning or significance, resulting in the inability to live independently. Cognitive impairment can influence a person’s mental flexibility, concentration, visual attention, and focused attention.

Trained professionals mainly administer commonly available neuropsychological tests, such as Mini-Mental State Examination (MMSE) [12], Montreal Cognitive Assessment (MoCA) [13], and Trail-Making Test (TMT) [14], to detect dementia. The traditional tests usually follow the pen-and-paper method. Patients experience psychological stress as a result of having to respond to a succession of queries for a longer period of time. If the patient has a problem with writing, this can affect the score generated with traditional screening methods. Eye-tracking technology has gained a significant role in screening cognitive and neurological problems [15]. Eye-tracking technology, which can monitor eye movements in a less intrusive way, helps to tackle this situation by assessing cognitive decline [16].

It is essential to identify people with signs of cognitive impairment and ensure their care and treatment by healthcare professionals. Considering the significant role of eye-tracking technology in determining cognitive problems, we propose a screening tool for the Eye-Tracking-Based Trail-Making Test (ETMT) to support healthcare professionals [8].

The objective of this study is to propose and develop the ETMT model, an Eye-Tracking-Based Trail-Making Test model that utilizes fuzzy-inference systems. The study aims to extract novel high-level features based on eye-tracking measures and generate fuzzy rules for the detection of various deficits contributing to cognitive impairment. The model further aims to provide detailed scores in visual search speed and focused attention, and an overall cognitive-impairment score. The primary goal is to offer psychologists valuable support in evaluating cognitive abilities by enhancing the tool’s ability to provide comprehensive cognitive assessments.

## 2. Related Works

The study on the related works is organized as, an overview of conventional methods used in cognitive impairment detection, exploring the significant role of eye tracking in identifying cognitive impairment, investigating cognitive impairment related to specific diseases, and highlighting the importance of the Trail Making Test (TMT) in detecting cognitive impairment.

### 2.1. Conventional Methods

There are numerous ways to identify a person’s cognitive impairment. Various neuropsychological tests are conducted with the assistance of trained healthcare professionals. MMSE [12], TMT [14], Alzheimer’s Disease Assessment Scale-Cognitive Subscale (ADAS-Cog) [7], and Frontal Assessment Battery (FAB) [7] are some of the frequently used tests to diagnose cognitive impairment and should be carried out by professional evaluators. While these neuropsychological tests are valid and trustworthy, they are not simple or brief enough to be utilized as regular dementia screening tools. MMSE [17] and MoCA [13] are some of the clinical tools to assess cognitive impairment. But they have limitations in identifying mild cognitive impairment. These scores can also be affected by age and education level.

Traditional methods use self-reporting questionnaires to detect cognitive decline in a person. There are well-validated and standardized questionnaires to detect cognitive decline, but they can be psychologically biased. Memory lapse, carelessness, or if the respondent voluntarily changes the feedback can affect the detection of cognitive changes.

Despite being accurate and reliable, these conventional cognitive tests have certain limitations. Senior patients may take a longer time to complete these tests. A more precise evaluation is required to detect the severity of cognitive impairment. Since they must respond to several questions during the examination, subjects may feel psychological stress. Since the proficiency of the assessor can influence the outcomes, these tests must be correctly administered by professional neuropsychologists with extensive training. Since some of these cognitive tests include writing and drawing, patients with motor dysfunction, which is frequently present in people with dementia, can impact the results.

### 2.2. Eye-Tracking Techniques

Eye-tracking data have a significant role in capturing the involuntary physiological responses of a person, which accurately helps to understand the characteristics or cognitive inhibitions of an individual [18]. Clinical assessments may miss identifying cognitive inhibitions in the earlier stage of disease, whereas simple eye-tracking measures can point that out [5,19].

Eye-tracking measures can distinguish among subtypes of mild cognitive impairment [19]. The antisaccadic task’s error rate was considered a significant measure to classify the types of mild cognitive impairment. Digit Span [20] and Spatial [21] Tests were performed on participants in the 55–90 age group. The analysis of variance (ANOVA) test could significantly point to the difference in the error rates in the antisaccadic tasks performed by different types of mildly cognitively impaired participants.

In a study performed with several memory tasks, the eye-tracking based score correlated with the MMSE score and could discriminate the control group from the Alzheimer’s disease (AD) and MCI groups [12]. Gaze tracking is widely accepted as a tool for monitoring a person’s cognitive functions and neurological disorders [22].

Eye-tracking technology can be utilized to assess and monitor various aspects of diseases like amyotrophic lateral sclerosis (ALS), AD, Parkinson’s disease (PD) [23], multiple sclerosis (MS), and epilepsy [24]. Eye tracking can detect early cognitive impairment by observing how individuals perform tasks such as looking away from sudden stimuli (antisaccadic task), smoothly tracking moving targets (smooth-pursuit task), and efficiently scanning visual scenes (visual-scanning task). Difficulty or errors in these tasks can indicate early changes in inhibitory control, motor control, attention, and scanning abilities, providing valuable insights into cognitive decline before conventional evaluations can detect them [19].

### 2.3. Cognitive Impairment Associated with Diseases

ALS patients exhibit cognitive and behavioral deficits due to their motor impairment. A higher antisaccadic error rate and saccadic latency were observed in these patients. In the successive stages of this disease, the patient loses the ability to write and speak, making the conventional paper–pencil method an inappropriate cognitive assessment tool [24]. The Edinburgh Cognitive and Behavioral ALS Screen (ECAS) is a standardized method to assess ALS patients [25]. The ECAS was developed exclusively for ALS patients and may not be appropriate for measuring cognitive and behavioral changes in people with other neurological disorders. The ECAS does not assess all aspects of cognitive function and is not sensitive to changes in cognitive function in people with early-stage ALS [26].

Lower motor neuron atrophy causes patients to lose their capacity to talk or write; at this point, these conventional measures are no longer appropriate for cognitive assessments [27]. An eye-tracking variant of the ECAS test could reduce the test duration and improve the evaluation effectiveness. Several studies show that eye tracking is a timely, efficient, and accurate way to evaluate cognitive performance in ALS patients [28,29].

Progressive memory loss is observed in AD patients and leads to dementia [30]. The saccades, fixations, and smooth pursuit performed by a patient could yield more accurate inferences of the stage of the disease.AD patients are slower in fixating on a target and have reduced fixation spans and less precise saccadic motions [24]. Gradual loss of attention and deterioration in visual attention were observed in AD patients [31].

While performing memory tasks, impaired visual attention and diminished visual interest were observed in AD patients. The memory-and-recall task [32,33], deductive reasoning [12], the working memory task [7], etc., helped to discriminate AD from the control group. Visual working memory tasks can also differentiate AD from MCI. A deductive reasoning task can distinguish the MCI group from the normal control group. The Alzheimer’s Disease Assessment Scale-Cognitive Subscale (ADAS-Cog) and the MMSE [34] are some of the established standard tests used to detect Alzheimer’s disease [35]. But these tests take a longer time to complete, and subjects may feel highly stressed due to the pressure of responding to a series of questions.

PD is a neurodegenerative disease that results in the patient’s cognitive decline [36]. In addition, ocular abnormalities and a longer response time were observed while performing saccadic tasks [24]. Degenerative changes can cause the impairment of focused attention. A decline in focused attention was observed in PD patients [37]. The visual search behavior of a PD patient is impaired, and there can be changes in fixational and non-fixational eye movements [38].

Standard conventional assessments like Parkinson’s Disease-Cognition (SCOPA-COG) and the Parkinson’s Disease-Cognitive Rating Scale (PD-CRS) [39] are not very sensitive in detecting cognitive abnormalities in the initial phases of PD. Eye-tracking measures are correlated with Parkinson’s disease severity, indicating that they may be used to forecast disease development in patients.

Disengagement of visual attention was noticed in people with cognitive impairment in their older age [31]. ’Gap’ and ’overlap’ conditions were introduced while displaying the stimulus to understand the saccadic movements of the participants. A blank screen was introduced between the fixation and target displays under the first condition, whereas an overlap of fixation and target was introduced between the fixation and target displays. Prosaccadic errors were observed in participants with dementia and old age.

The memory test visual paired-comparison (VPC) task is used to understand memory impairment [40]. Eye movement features like fixations, saccades, and re-fixations were considered to understand the participants’ behavior while viewing novel and repeated stimuli. The machine learning algorithm support vector machine (SVM) could accurately distinguish the normal control group from the MCI group. The VPC task with eye-tracking technology was used as a screening tool to detect MCI [41]. Eye-tracking features like total looking time, fixation count, and percentage of time spent on the novel image were considered for analysis. The novel image was viewed by the control group and PD patients for more than 70% of their total viewing time but only for 53% of the time by the MCI group.

A deficit in sustained attention is observed in the early stage of dementia and leads to severe impairment in the later stage [42]. An earlier diagnosis helps to slow down the disease’s growth in a person with MCI [43,44,45]. But, unfortunately, the clinical biomarkers to detect these diseases earlier are quite expensive and invasive [19]. Eye tracking provides non-invasive and involuntary measures that serve as a critical biomarker in the earlier diagnosis of these diseases. Advanced neurophysiological eye-tracking measures are gaining researchers’ attention, considering the low responsiveness of conventional approaches to identify cognitive abnormalities in the initial phases of disorders [24].

It is challenging to detect MCI in an early stage based on clinical evaluations [46]. However, the eye-tracking measures [47] captured while performing the working memory task can distinguish healthy individuals from people with cognitive impairment. When conventional assessment methods fail to detect minute impairment, simple eye-tracking techniques can pinpoint that impairment. Various tasks can be used to understand the different types of cognitive-impairment diseases. Different types of tests are performed to understand different cognitive-impairment diseases [43].

The myelin sheath, optic neurons, and the spinal cord are all harmed by the disease MS [24]. The impairment of memory, executive function, and attention is increasingly recognized as a substantial functional weakness in MS. Neurophysiological examinations and brain MRI have traditionally been adopted as diagnostic tools. Using eye tracking, abnormal visuospatial behavior in MS can be detected. The most popular method for evaluating oculomotor function in MS patients is the use of saccadic tasks because of their direct relationship with ocular-nerve damage.

Cognitive impairment associated with stroke includes deficits in attention, memory, language, and executive functions [48]. These impairment types significantly impact the quality of life in stroke survivors, and there is a high potential for the development of dementia within the first year of stroke onset. After a stroke, cognitive impairment, including memory problems, can be detected with neuropsychological assessments such as MMSE, MoCA, and TMT, and language tests like phonological and semantic fluency token tests. An eye-tracking-based study was performed in stroke patients with cognitive impairment [49]. They underwent eye-tracking linkage attention training and showed significant improvements in visuospatial and memory tests, suggesting that eye-tracking measures can detect cognitive impairment and contribute to its rehabilitation in stroke patients.

People with Huntington’s disease (HD) can experience cognitive decline that makes it difficult for them to think, remember, and learn. This can happen even before the motor symptoms of HD appear [50]. Researchers found that premanifest (Pre-HD) people made more errors and took longer to respond when the task required inhibitory control, working memory, or fronto-executive function [51]. People with HD have lower saccadic latency and are more likely to make disinhibited saccades [52]. Eye tracking is a promising tool for detecting changes in saccadic eye movements, which could serve as biomarkers for tracking Huntington’s disease (HD) progression.

The various diseases associated with cognitive impairment, the standard assessment tools used to detect impairment and these tools’ shortcomings, and the eye-tracking measures used for impairment detection are shown in Table 1. The various tasks performed to screen mild cognitive impairment are memory tasks [7], attention and calculation tasks [7], TMT-A, TMT-B [14], the Word Memory Test [53], identifying some objects, and counting backward [54]. People with AD progressively lose their ability to exert effective inhibitory control over their actions and their attention. In particular, the capacity to control and inhibit voluntary gaze shifting away and towards the salient stimulus is impaired. Physicians have a significant opportunity with eye tracking to diagnose Alzheimer’s disease [55] in its earliest stages during the MCI phase [56].

### 2.4. Trail-Making Test

The TMT is a neuropsychological test [58] that measures the capability of the brain to perform visual attention and task switching [59]. It provides details regarding executive functioning, visual search speed, scanning, processing speed, focused attention, and mental flexibility. Many types of brain dysfunction, especially those affecting the frontal lobes, can be detected and diagnosed using the TMT. This area of the brain regulates high-level cognitive skills, such as planning, consciousness, memory, emotion, and attention. It was used as a standard test to evaluate soldiers’ brain damage during the Second World War. The test was administered using paper and a pencil in a conventional manner. To identify cognitive impairment, only the error rate and overall completion time were taken into consideration and could not offer a detailed analysis [60]. The TMT’s objective is to carry out the examinations as promptly and precisely as possible in order to identify any potential indicators of cognitive impairment. A psychologist should assist the patient throughout the conduct of the test. People who suffer from motor impairment may have difficulty in completing the test and take longer to complete it. There have been studies on different TMT variants to overcome the limitations of existing conventional methods.

The digital TMT (dTMT) precisely monitors a variety of distinct elements along with the overall completion time and the number of errors, such as the number of viewing pauses, the duration of each pause, lifting, lifting duration, duration within the circle, and the amount of time across the circles [61].

The benefits of combining infrared eye tracking with the TMT task is another significant factor [62]. Research on infrared eye tracking is becoming well known for its ability to diagnose cognitive impairment. Eye tracking was employed in an ongoing study to gather objective, quantitative data on each participant’s visual, attentional, and memory functions [63]. In addition to the basic task-completion-time metric, eye tracking enabled a number of potentially insightful and sensitive measurements.

### 2.5. Summary

The study of related works revealed a variety of cognitive deficits, including memory loss, lower visual interest, atypical visuospatial behavior, attentional disengagement, motor impairment, and impaired mobility, and their association with diseases like AD, PD, MCI, MS, ALS, HD, stroke, and epilepsy. Eye-tracking features like inattentional blindness, error rate, total completion time, scanpath comparison score, fixation duration, saccadic latency, and smooth pursuit could allow for the drawing of inferences to understand those deficits, as shown in Table 2. The eye-tracking version of the TMT provides those eye-tracking features and can allow for the drawing of inferences on detecting cognitive impairment.

In a previous study, [1], a comparison of various versions of the TMT that are performed digitally, using paper–pencil, or with eye tracking was reported. The benefits of the eye-tracking version of the TMT were also pointed out based on our findings. The study presented individual and group profiles, which aids in comprehending a group of individuals performing the TMT with a similar level of cognitive decline [1].

The proposed ETMT model is a screening tool for cognitive impairment based on the changes observed in eye-tracking measures while performing eye-tracking versions of TMT-A and TMT-B. The ETMT model provides a detailed understanding of the participant’s focused attention and visual search speed other than their cognitive impairment. By utilizing eye-tracking technology, the model enhances practitioners’ understanding of these cognitive aspects, enabling a more detailed evaluation of participants’ cognitive abilities and potential deficits.

## 3. Materials and Methods

The system architecture of the ETMT model is shown in Figure 1. Eye-tracking data are collected based on the TMT stimulus. Low-level, middle-level, and high-level features are extracted from the raw eye-tracking data. The extracted features supplied to the ETMT model provide three scores: visual search speed, focused attention, and overall cognitive impairment. Two separate fuzzy-inference systems generate the visual-search-speed and focused-attention scores. An adaptive neuro-fuzzy-inference system (ANFIS) based on all the extracted features generates the overall cognitive-impairment score. The detailed architecture of the ETMT model is shown in Figure 2. The following sections explain each module of the ETMT model.

The cognitive-impairment score can only provide insights into a patient’s overall cognitive functioning. But the scores in visual search speed and focused attention provided by the ETMT model serve as sensitive markers and indicate deficits such as processing speed deficit, visual attention deficit, motor impairment, attentional disengagement, memory deficits, neuropsychological impairment, and executive functioning. These scores serve as sensitive markers and point to the exact cognitive deficit of the patient. Each disease associated with cognitive impairment may have different types of deficits. The detailed score can provide an earlier diagnosis and guide the psychologist to start the appropriate treatment that helps to delay the advancement of the disease. Furthermore, the use of eye tracking enhances the feedback that practitioners receive from the TMT by identifying suboptimal aspects of cognition.

### 3.1. Data Collection

This study was performed on participants working in a hospital (mean age = 30.6, SD = 8.6; age range = 20 to 54). The demographic details of the participants are shown in Figure 3. The eye tracker used for data collection was an SMI REDn Professional eye tracker. The sampling frequency of the eye tracker was 60 Hz. Exclusion criteria involved participants with drooping eyelids, contact lenses, squinting, difficult glasses, glasses with more than one power (bifocals, trifocals, and progressives), the use of alcohol or psychotropic medication, recent eye surgery, and eye alignment abnormalities [64]. The participants who did not meet the specified exclusion criteria were included in the study. A total of 40 participants were recruited for the study, and we collected data from those participants. Out of the collected data, 9 participants’ data were discarded due to technical issues and data loss, leaving a total of 31 participants’ data for analysis. The flow of the conduct of the experiment is shown in Figure 4. The pre-survey procedure included obtaining their consent and collecting their demographic details. The guidelines were explained to the participant before obtaining their consent. After obtaining their consent, their demographic details were collected. Next, participants were asked to sit in a comfortable position and restrict head movements during the experiment. The distance between the screen and the participant was always maintained between 50 and 60 cm [65]. The stimulus was displayed with the help of Experimental Suite software. Calibration was performed at the start of each experiment. Nine-point calibration was performed to understand the characteristics of the participant’s eye. The eye-tracking version of the TMT, considered the screening test for cognitive impairment, was used as the stimulus [61,66]. The stimulus was displayed in the order of TMT-A simple, TMT-A complex, TMT-B simple, and TMT-B complex. TMT-A is the stimulus with number combinations, and TMT-B is the stimulus with number and alphabet-letter combinations. TMT-A simple is a simple random number sequence from 1 to 8. TMT-A complex is a random assortment of numbers between 1 and 25. TMT-B simple is a random combination of alphabet letters from A to D and numbers from 1 to 4. TMT-B complex is a random combination of alphabet letters from A to L and numbers from 1 to 12 [67]. The participants were instructed to look at the ascending numbers in TMT-A and the ascending combination of numbers and alphabet letters in TMT-B. While watching them, the participants were instructed to speak out aloud the number or alphabet letter. The individual was then asked to complete the conventional TMT after the eye-tracking version. It is based on the paper–pencil method, and participants were instructed to link the alphabet letters or numerals in an ascending sequence. The eye tracking and traditional TMTs followed the same order of stimuli.

### 3.2. Feature Extraction

The raw data obtained had features like timestamp, type of stimulus, gaze details, and pupil diameter details. Experimental Suite software 3.7 was used to extract low-level and middle-level features. The low-level features were extracted based on fixations, saccades, and blinks. The ETMT model considers the basic features extracted from fixations as low-level features FL1 (Feature Low-Level 1) to FL4 (Feature Low-Level 4). Middle-level features FM1 (Feature Middle-Level 1) to FM5 (Feature Middle-Level 5) were extracted based on each specific area of interest (AOI). An AOI is a user-defined sub-region in a stimulus that helps to extract the metrics specific to those regions. Each number and alphabet letter in the stimulus is defined as an AOI. The fixation-based low-level, middle-level, and high-level features used in the ETMT model are shown in Table 3. High-level features FH1 (Feature High-Level 1) to FH4 (Feature High-Level 4) were extracted from low-level and middle-level features by applying specific algorithms, as shown in Algorithms 1–3. The feature scanpath score (FH1) was extracted based on Algorithm 1, which generates a scanpath string [68]. A scanpath is a visualization that shows the sequence of fixations and saccades in the order in which it is visited [1]. The scanpath string specifies the sequence of fixations visited in order. The Levenshtein distance was calculated between each participant’s scanpath string and the expected scanpath string. The expected scanpath string is the expected order of viewing the AOIs in the stimulus. The calculated Levenshtein distance is considered the scanpath score [69]. The overall amount of time required to complete the entire task is the feature total time (FH2). The feature error rate (FH3) is the count of missed targets. While viewing the AOIs in the specified order, there can be fixations outside the AOIs and wrong fixations. So, the total count of those errors indicates the error rate. Inattentional blindness (FH4) occurs when a person fails to notice something completely visible to them [70]. It indicates the attentional abilities and working memory of a person [71]. Lower working memory is observed in people with inattentional blindness [72].

**Algorithm 1** Scanpath string generation Define the AOIs Get the fixations i←1 j←1 N1←count_fixation N2←count_AOIs **while** i≠ N1 **do**     **while** j≠ N2 **do**         **if** Fixation falls within AOI **then**              Print AOI Name         **end if**     **end while** **end while**

**Algorithm 2** Error rate Define the AOIs Get the fixations error_rate←1 i←1 j←1 N1←count_fixation N2←count_AOIs **while** i≠ N2 **do**     **while** j≠ N1 **do**         **if** Fixation not falls within AOI **then**              error_rate+=1         **end if**     **end while** **end while** Print error_rate

**Algorithm 3** Inattentional blindness**Require:** Scanpath String**Ensure:** Inattentional Blindness Score **function** CalculateInattentionalBlindness(scanpath)     count←0     i←0     N←length(scanpath)     **while** i<N **do**         **if** repetition of a pattern in the scanpath starting from index *i* **then**             count←count+1             i←i+length(pattern)            ▹ Skip the repeated pattern         **else**              i←i+1         **end if**     **end while**     **return** count **end function**

### 3.3. Visual-Search-Speed Fuzzy-Inference System

The specific score in visual search speed of a participant can lead to inferring the exact deficit of the participant. The TMT follows a visually guided task where the participants need to visually attend each AOI in a specific order [73]. The visual search pattern of an individual may differ from that of others [74]. The visual-search-speed score indicates the efficiency in visual search and speed of completion. The gaze patterns of an individual are the indicators of their visual behavior [75,76]. An individual’s cognitive state and activities can be discovered based on their visual behavior [77].

The visual-search-speed fuzzy-inference system (FIS) takes high-level features FH1 and FH2 as the inputs and generates a score based on the generated rules [65]. FH1 is generated based on the comparison with the scanpath that correctly follows the path necessary to complete the given task [78]. FH2 is the time required to complete the entire task. The completion time indicates the speed and focuses on identifying each AOI. The Mamdani model for the visual-search-speed FIS is shown in Figure 5. The parameters of the visual-search-speed FIS are given in Table 4.

By utilizing this visual-search-speed score, the fuzzy-inference system can provide valuable insights into various deficits, particularly those related to visual attention and processing speed. An individual’s visual attentional efficiency and effectiveness may be reflected in their visual-search-speed score. A person with deficits in visual attention may exhibit slower scanpath scores (FH1) compared with the reference path [79]. This could indicate difficulties in properly attending to and processing relevant visual stimuli within the task. Processing speed refers to how individuals perceive, analyze, and respond to visual information. In the fuzzy-inference system context, people with processing speed deficits might exhibit longer total completion times (FH2) than those without deficits. A slower completion time indicates difficulty in processing visual information effectively and quickly enough to make decisions. The visual-search-speed FIS provides an indication of deficits in processing speed and visual attention.

### 3.4. Focused-Attention Fuzzy-Inference System

The capacity to concentrate on a task without getting distracted is known as focused attention. Maintaining a constant behavioral response throughout an ongoing activity is known as sustained attention. The focused-attention fuzzy-inference system (FIS) considers high-level features FH3 and FH4 as inputs and generates a score based on the generated rules. FH3, the error rate, and FH4, inattentional blindness, contribute to detecting focused attention. The Mamdani model for the focused-attention FIS is shown in Figure 6. The parameters of the focused-attention FIS are given in Table 4.

The focused-attention score obtained with the fuzzy-inference system can provide valuable information regarding several deficits, including motor impairment, attentional disengagement, memory deficits, neuropsychological impairment, and executive functioning. The focused-attention score can indirectly reflect motor impairment by considering the error rate as an input. Higher error rates indicate challenges in accurately performing motor tasks and can show signs of underlying motor impairment. Attentional disengagement refers to the difficulty in shifting attention from the current task to a different task or stimulus. Higher levels of inattentional blindness can indicate a deficit in attentional disengagement, as people can find it difficult to shift their focus away from the main task in order to observe relevant unexpected stimuli.

Memory deficits involve difficulty in storing or retrieving information. The focused-attention score can indirectly provide insights into memory deficits by considering the error rate and inattentional blindness as inputs. Higher error rates and increased inattentional blindness may indicate difficulty in storing and retrieving task-related information, indicating potential memory deficits. Neuropsychological impairment encompasses a range of cognitive deficits resulting from neurological conditions or brain injuries. The focused-attention score can indicate such impairment by considering the overall performance on the task, as reflected in the error rate and inattentional blindness inputs. A lower focused-attention score may suggest the presence of neuropsychological impairment. Executive functioning refers to a set of cognitive processes responsible for goal-directed behaviors, such as planning, problem solving, and cognitive flexibility. The focused-attention score can indirectly provide insights into executive functioning deficits by considering error rate and inattentional blindness. Higher error rates and increased inattentional blindness may indicate challenges in executing efficient executive functions. Considering all these deficits, the focused-attention FIS indicates motor impairment, attentional disengagement, memory deficits, neuropsychological impairment, and executive functioning [80].

### 3.5. Adaptive Neuro-Fuzzy-Inference System (ANFIS)

All the extracted features were considered for generating the overall cognitive-impairment score. First, the k-means clustering algorithm was applied to label the data by considering k = 3. All the extracted features of all the 31 participants were given as inputs to the clustering module, which clustered the entire samples into three groups based on the similarities of the data. Then, the clustered data were labeled based on expert knowledge of data. Finally, the clustered data were fed into the ANFIS, which automatically generated the rules and score.

The ANFIS is a hybrid model that combines an adaptive artificial neural network (ANN) and the FIS [81,82]. Since it combines both an adaptive ANN and the FIS, it has the benefits of both. It corresponds to a fuzzy model of Takagi–Sugeno that generates the fuzzy IF-THEN rules according to the relationship between inputs and outputs. It employs a hybrid learning algorithm that combines the backpropagation technique and the least squares approach.

The ANFIS has five layers. The fuzzy layer conducts the fuzzification of the inputs; the product layer calculates the firing strength of a rule using multiplication; the normalization layer normalizes the fuzzy strengths from the previous layer; the de-fuzzy layer conducts the de-fuzzification of the inputs and the output layer, which is a single node that finds the sum of incoming signals.

The four low-level features, five middle-level features, and four high-level features of all 31 participants were labeled and provided as inputs to the ANFIS model. The ANFIS architecture for the detection of the overall cognitive-impairment score is shown in Figure 7. The parameters for training the ANFIS model are shown in Table 5.

The cognitive-impairment score can only provide insights into a patient’s overall cognitive functioning. But the scores in visual search speed and focused attention provided by the ETMT model serve as sensitive markers and indicate deficits such as processing speed deficit, visual attention deficit, motor impairment, attentional disengagement, memory deficits, neuropsychological impairment, and executive functioning. These scores serve as sensitive markers and point to the exact cognitive deficit of the patient. Each disease associated with cognitive impairment may have different types of deficits. The detailed score can provide an earlier diagnosis and guide the psychologist to start the appropriate treatment that helps to delay the advancement of the disease. Furthermore, the use of eye tracking enhances the feedback that practitioners receive from the TMT by identifying suboptimal aspects of cognition.

## 4. Result Analysis

The ETMT is a screening tool for detecting cognitive impairment. It generates scores based on visual search speed, focused attention, and overall cognitive impairment. Two fuzzy-inference systems are modeled to generate scores based on visual search speed and focused attention. Finally, the overall cognitive-impairment score is generated using the ANFIS model considering all the extracted features. The FIS for generating the visual-search-speed score considers FH1 and FH2 as inputs, and for focused attention, it considers FH3 and FH4 as inputs.

The results of Participant 1 with high visual-search-speed score and focused-attention score are shown in Figure 8. A participant with lower scanpath scores and less time needed to complete the task has a high visual-search-speed score [83]. A participant with a lower error rate in completing the task and the absence of inattentional blindness has great focused attention. The scanpath shown in Figure 8 indicates that Participant 1 had a lower error rate and no inattentional blindness.

The results of Participant 2 with medium visual-search-speed score and focused-attention score are shown in Figure 9. A participant with medium scanpath scores and medium time needed to complete the task has a medium visual-search-speed score. A participant with a medium error rate in completing the task and the presence of inattentional blindness has medium focused attention. The scanpath shown in Figure 9 indicates that the participant had a medium error rate. Without noticing the visits to AOIs 6 and 7, the participant continued the search and repeated visiting those AOIs. The repeated visits show that the participant had inattentional blindness. Individuals with higher cognitive impairment may be more susceptible to inattentional blindness due to difficulty in allocating and maintaining attention. Their cognitive impairment, such as deficits in attention, working memory, or executive functions, may hinder their ability to detect and process unexpected stimuli even when they are directly in their visual field.

The ANFIS model, which considers all the extracted features, generates the overall cognitive-impairment score. A total of 25% of the sample was used for testing, while the remaining 75% was used to train the model. The ANFIS model was trained for an epoch number of 20 to 50, and the error statistics were analyzed. A lower root mean square error (RMSE) was observed for an epoch number of 30. The error statistics of the ANFIS model are shown in Table 6.

Each individual receives a comprehensive assessment that includes a detailed visual-search-speed score, a focused-attention score, a cognitive-impairment score, and the corresponding indication of specific deficits. Figure 10 shows the overall score generated for Participant 1. The figure shows the scanpath generated by Participant 1 while performing the TMT task. They generated a scanpath string of ’6 1 2 3 4 5 6 7 8’ with a scanpath score of 1. A lower scanpath score, along with a shorter completion time, indicates a higher visual-search-speed score. The scanpath of Participant 1 indicated a single error in task performance and the absence of inattentional blindness, which demonstrates a high level of focused attention. High visual-search-speed score and focused-attention score contribute to a lower cognitive-impairment score. Upon detailed examination of the visual-search-speed score, it was evident that Participant 1 exhibited minimal deficits in visual attention and processing speed. The focused-attention score provides an indication of deficits in motor impairment, attentional disengagement, memory deficits, neuropsychological impairment, and executive functioning. For Participant 1, all these deficits were low, indicating minimal impairment.

Figure 11 shows the scores of Participant 2. Participant 2 demonstrated moderate cognitive-impairment, visual-search-speed, and focused-attention scores. Their scanpath displayed an irregular and disorganized pattern compared with Participant 1, resulting in lower visual search speed. Multiple errors in task performance and potential instances of inattentional blindness suggest reduced focused attention compared with Participant 1. Participant 2 exhibited a small visual attention deficit but had a medium deficit in processing speed. The focused-attention score indicated moderate impairment across various areas. Compared with Participant 1, Participant 2 experienced a medium amount of cognitive impairment.

Each generated score was analyzed based on the age and gender of each participant, as shown in Figure 12 and Figure 13. It was observed that the younger age group had high or medium visual-search-speed scores and focused-attention scores. Same-age category people showed low or medium cognitive-impairment scores. The oldest age group showed a low visual-search-speed score. Low and medium focused-attention scores were observed in the old age group. High cognitive impairment was observed in the same age group. Based on the gender analysis, most female participants had a higher visual-search-speed score than male participants. No female participant was observed to have a low focused-attention score. Only one female participant was observed to have a high cognitive-impairment score.

The participants’ scores were analyzed, and it was observed that the participants with high visual search speed and focused attention had significantly lower cognitive-impairment scores. Most participants with high visual-search-speed and focused-attention scores had low or medium cognitive-impairment scores, and vice versa. The distribution of participants in each score is shown in Figure 14. The figure shows the count of participants falling into a specific category of the score. It clearly shows that the eight participants with high visual-search-speed scores and the eleven participants with high focused-attention scores fall into the category of low cognitive-impairment scores. Same way, four participants with low visual-search-speed scores and the four participants with low focused-attention scores fall into the category of high cognitive-impairment scores. Furthermore, the other numbers in each box specify the counts of participants falling into subsequent categories, providing a comprehensive overview of the classification outcomes across different cognitive-impairment levels.

Figure 15, Figure 16 and Figure 17 show the distribution of each feature based on the cognitive-impairment score. Clear differences among groups can be seen in the boxplot analysis of eye-tracking features based on low, medium, and high cognitive-impairment scores. These distinct patterns provide strong justification for considering eye-tracking measures to be reliable indicators of different levels of cognitive impairment. This indicates that eye-tracking features can capture meaningful differences in eye movement behavior associated with cognitive impairment and have the potential to serve as valuable tools for assessing cognitive impairment.

The clustering and labeling of the data were validated with machine learning algorithms like decision trees, linear discriminant analysis (LDA), neural networks, k-nearest neighbor (KNN), support vector machine (SVM), and naive Bayes. In total, 75% of the sample was used for training, and 25% of the data were used for testing. The Matlab toolbox ’classification learner’ was used for analysis. The results obtained are shown in Table 7. The data were also collected using the traditional TMT to validate the ETMT model. The completion time was used as the criterion for evaluation, and it was examined how the traditional and eye-tracking versions of the TMT correlated with one another. Based on the value of correlation coefficient *p*, there is sufficient evidence to conclude that there is a linear relationship between the time of completion in the eye-tracking and traditional versions of the TMT. The correlation score and the value of *p* are shown in Table 8.

## 5. Discussion

The ETMT model, TMT’s eye-tracking variant, can address the impairment associated with focused attention and visual search speed by providing specific scores in those deficits. The model has the capability to extract specific high-level features, denoted as FH1 to FH4, which play a crucial role in detecting cognitive impairment. The significance of features such as scanpath comparison score, total time for task completion, error rate, and inattentional blindness lies in their ability to contribute to the detection of cognitive impairment in individuals. The scanpath comparison score helps assess the coherence and efficiency of eye movements during visual tasks, offering valuable information about attentional focus and cognitive processing. The total time for task completion reflects processing speed and efficiency, which can be compromised in individuals with cognitive impairment. The error rate provides insights into the accuracy of cognitive processing, with higher error rates indicating potential cognitive deficits. Lastly, inattentional blindness, the failure to notice salient stimuli, can indicate attentional impairment and cognitive difficulties. Collectively, these features serve as important indicators that could aid psychologists in identifying cognitive impairment and developing appropriate intervention strategies.

Standard tests like ECAS, MMSE, MoCA, and ADAS-CoG detect specific diseases by identifying the specific impairment type associated with those diseases. The impairment types that can be identified with standard tests are shown in Table 9. Compared with other tests, the ETMT model allows for the drawing of more accurate inferences to identify various impairment types. The ETMT can reveal information about an individual’s cognitive performance, depending on how rapidly a person can search, assess, and interpret visual inputs without getting distracted. This assessment also reveals a person’s level of mental flexibility, focused attention, and processing speed. All of these skills fall under the umbrella of executive functioning. A significant loss or reduction in these skills could be a sign of cognitive impairment.

As shown in Table 10, several shortcomings are associated with standard methods. The ETMT model, the eye-tracking version of the TMT, can overcome many of those shortcomings. One of the crucial challenges for conventional methods is the need for a trained evaluator throughout the conduct of the tests. Also, many of the tests take a longer time to complete. A study by the Observer Research Foundation (ORF) in 2021 reported a shortage of mental health professionals [84]. Most of the time, crowded clinics and residential settings make it impossible to use the present methods to detect decreased motor and cognitive performance. The advantage of the ETMT model over other models is that no trained evaluator is needed to administer the test. The tests can also be finished quickly by the participants. While comparing the ETMT with the traditional TMT, we observed a correlation in completion time. The minimum time taken by a participant to complete the entire ETMT test was 67 s, and the maximum was 168 s. The traditional TMT had a minimum completion time of 38 s and a maximum completion time of 184 s. The time duration for each task in ETMT and traditional TMT data collection is shown in Table 11. Participants were not under any stress during the experiment because the tests were simple and required little time. The ETMT model not only incorporates the benefits of the traditional TMT model, particularly in terms of completion time, but it also overcomes its limitation of having a trained evaluator by utilizing eye-tracking and fuzzy-inference techniques. It is inferred from the table that ETMT does not have any of the shortcomings compared to other methodologies.

Compared with standard cognitive assessments, eye-tracking technology can help assess cognitive impairment with higher accuracy and precision. Oculomotor information gathered while performing cognitive tasks can shed light on physiological systems. For patients with cognitive impairment, eye tracking offers a nonverbal and less cognitively exhausting way to monitor disease development. By examining individuals’ eye movement behavior, researchers could better understand how eye movement patterns are meaningfully translated into visual experiences while conducting the TMT task and how they are related to particular reactions. The ETMT model, a combination of eye tracking and the TMT, could profoundly change the conventional way of detecting cognitive impairment.

The ETMT model is limited to considering only features based on eye tracking based on the TMT for detecting cognitive impairment. The ETMT model does not take into account any additional characteristics based on facial expressions or observations of other physiological signals. In the future, the ETMT model could be further enhanced by incorporating additional provisions to identify and assess various types of impairment, ultimately enabling the generation of comprehensive recommendations tailored to the specific findings.

## 6. Conclusions

The ETMT is a screening tool that supports healthcare experts by providing valuable indicators of an individual’s cognitive impairment based on the TMT. Our main contribution was to extract significant high-level features for generating cognitive-impairment scores. The high-level features of error rate, scanpath comparison score, total time, and inattentional blindness were used to generate all three scores for screening cognitive impairment. Other than the overall cognitive-impairment score, the ETMT provided detailed scores in visual search speed and focused attention, which have significant roles in understanding the exact deficits of a patient. The ETMT tool facilitates an in-depth evaluation of an individual’s cognitive abilities, specifically in the domains of visual perception and attentional processes. The integration of these advanced computational techniques enables psychologists to obtain detailed and nuanced information about an individual’s cognitive functioning, leading to more accurate diagnoses, personalized treatment plans, and improved patient care. Moreover, the integration of eye-tracking technology enriches the feedback received by practitioners during the administration of the TMT, as it enables the identification of cognitive aspects that may not be functioning optimally. The ETMT model serves as a screening tool for cognitive impairment, provides a cost-effective diagnosis that considers the individual’s comfort, and allows for the reallocation of healthcare resources to patients with severe ailments.

The testing of the ETMT model was limited to samples that were not uniformly distributed based on age. Since the study did not involve actual patients, the presented results cannot be directly transferred to clinical diagnosis. However, the study demonstrated a detailed understanding of cognitive impairment by providing multiple scores in the participants’ deficits. Since the stimulus used for the study was the TMT, which gives significance to fixation-based features, there was a limited analysis based on saccadic features.

In the future, the ETMT could be integrated with eye-tracker software as a plugin tool and automatically screen an individual’s cognitive impairment. Saccadic and blink features could also be included in the ETMT model to understand other cognitive-impairment types. Actual patients’ data could help ensure the ETMT model’s efficiency. We aim to collect data from patients by considering other physiological measures as the ground truth in the future. The current ETMT model is limited to visual-search-speed score, focused-attention score, and overall cognitive-impairment score. Other impairment types could also be included in the future to detect scores and provide recommendations based on each impairment type or disease.

## Figures and Tables

**Figure 1 sensors-23-06848-f001:**
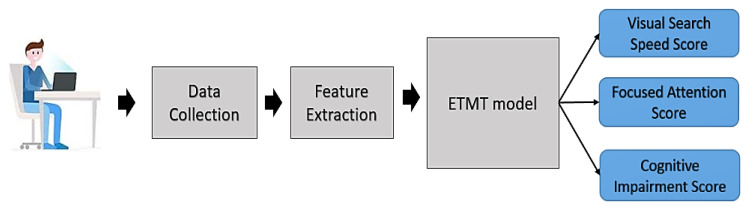
System architecture of ETMT model.

**Figure 2 sensors-23-06848-f002:**
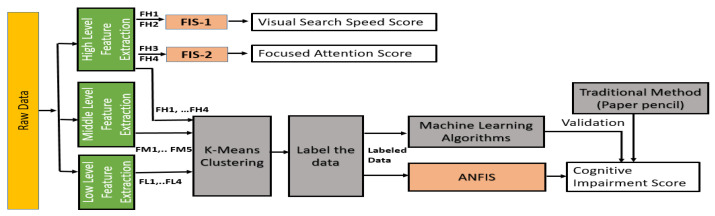
Detailed architecture of ETMT model.

**Figure 3 sensors-23-06848-f003:**
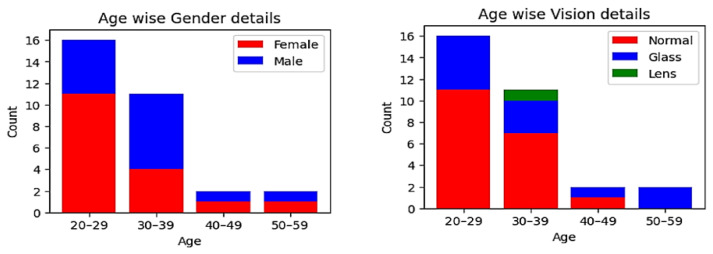
Demographic details.

**Figure 4 sensors-23-06848-f004:**
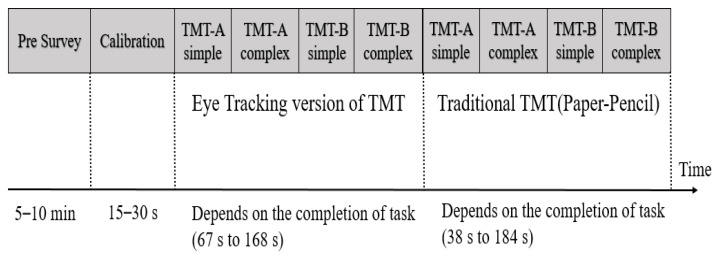
Experimental setup. TMT-A is Trail-Making Test with number combinations, TMT-B is Trail-Making Test with number and alphabet-letter combinations.

**Figure 5 sensors-23-06848-f005:**
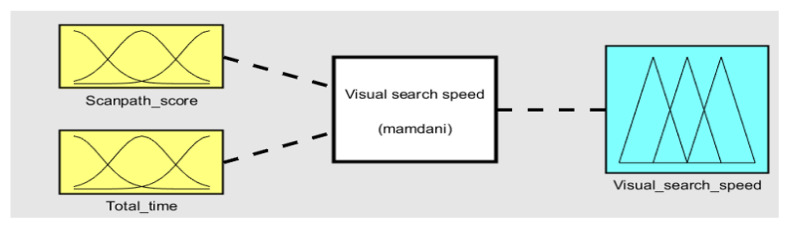
Visual-search-speed fuzzy-inference system. FIS-1 submodule in ETMT Model.

**Figure 6 sensors-23-06848-f006:**
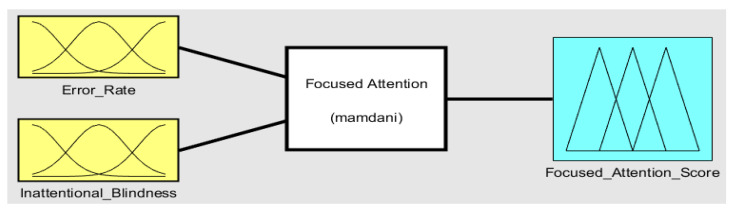
Focused-attention fuzzy-inference system. FIS-2 submodule in ETMT Model.

**Figure 7 sensors-23-06848-f007:**
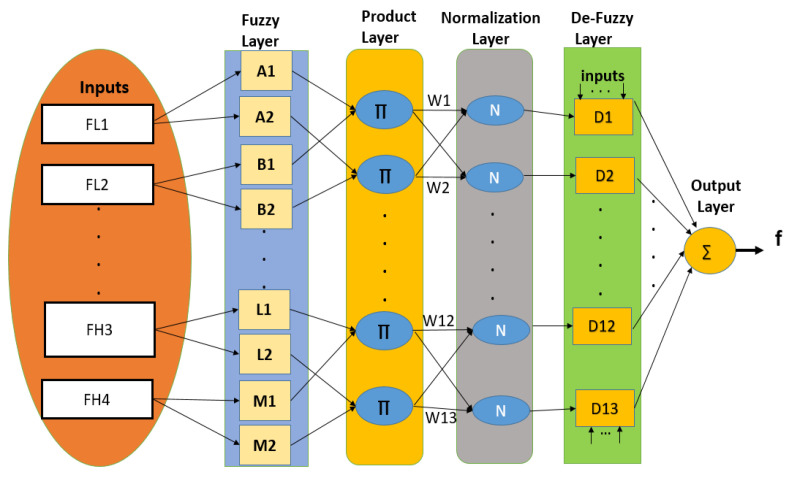
Architecture of ANFIS (submodule in ETMT Model) for overall cognitive-impairment score.

**Figure 8 sensors-23-06848-f008:**
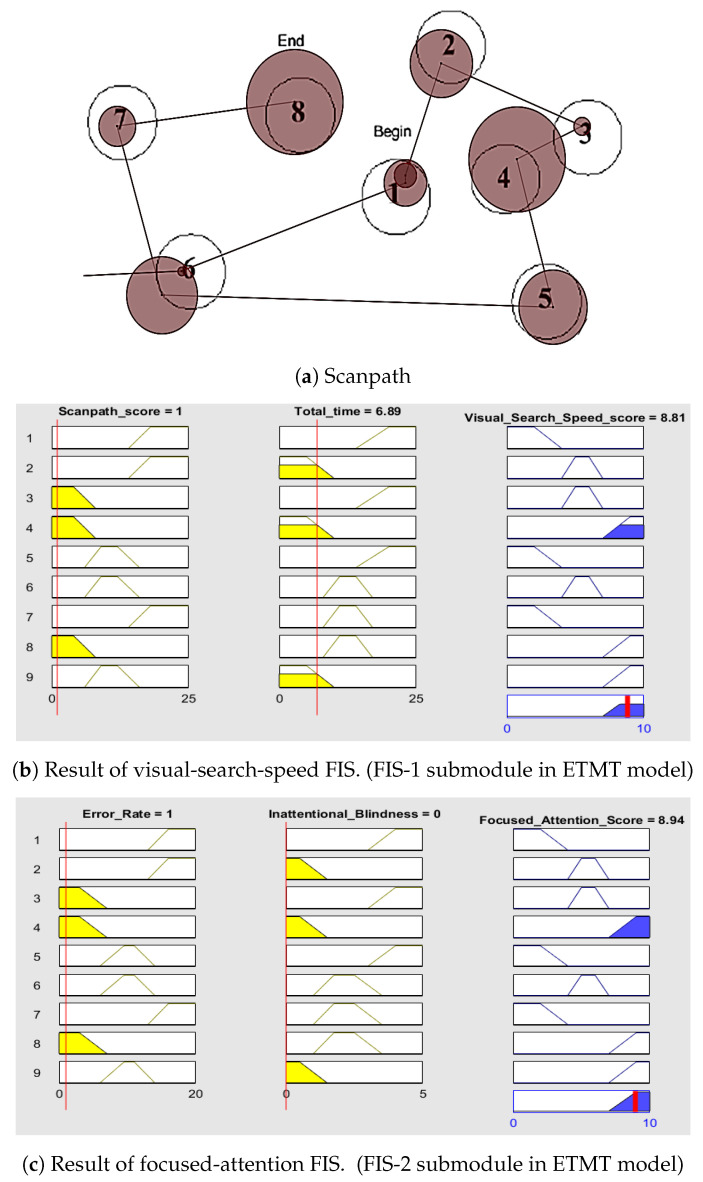
Scanpath and the result of FIS of participant with high visual-search-speed and focused-attention scores. Scanpath: Order of viewing the AOIs in the TMT-A stimulus. Red thin line: Membership function of the input variable. Yellow color: Fuzzy set to which the input variable belongs. Blue color: Fuzzy set to which defuzzified output belongs. Red thick line: Defuzzified output.

**Figure 9 sensors-23-06848-f009:**
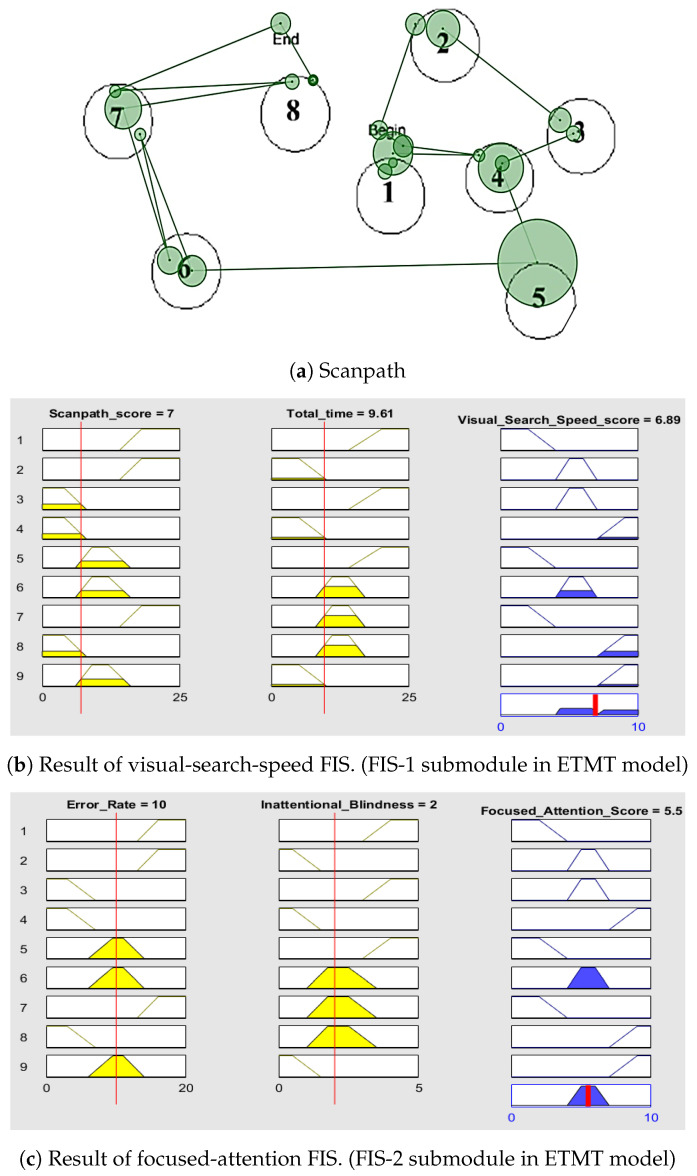
Scanpath and the result of FIS of participant with medium visual-search-speed and focused-attention scores. Scanpath: Order of viewing the AOIs in the TMT-A stimulus. Red thin line: Membership function of the input variable. Yellow color: Fuzzy set to which the input variable belongs. Blue color: Fuzzy set to which defuzzified output belongs. Red thick line: Defuzzified output.

**Figure 10 sensors-23-06848-f010:**
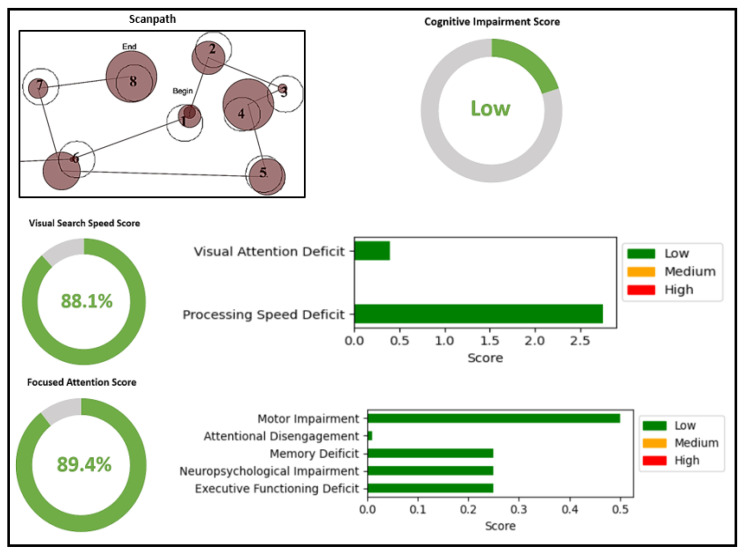
Detailed scores generated with ETMT model for a person with low cognitive impairment. Scanpath shows the order of viewing the AOIs in the TMT-A stimulus.

**Figure 11 sensors-23-06848-f011:**
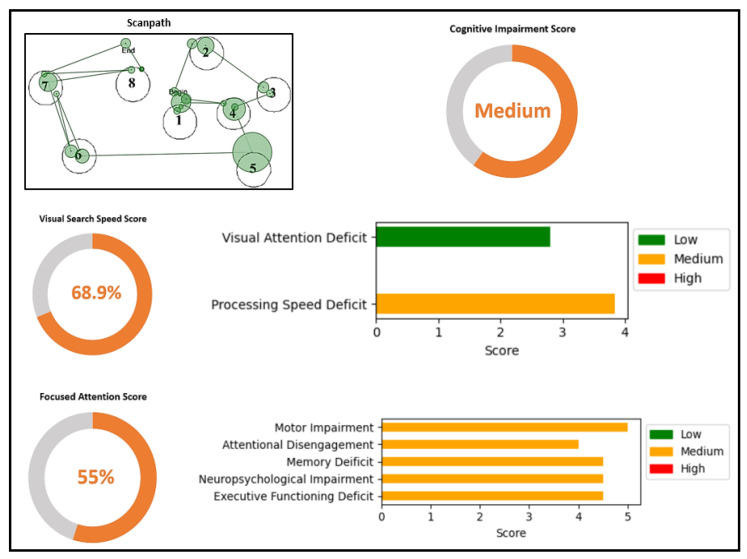
Detailed scores generated with ETMT model for a person with medium cognitive impairment. Scanpath shows the order of viewing the AOIs in the TMT-A stimulus.

**Figure 12 sensors-23-06848-f012:**
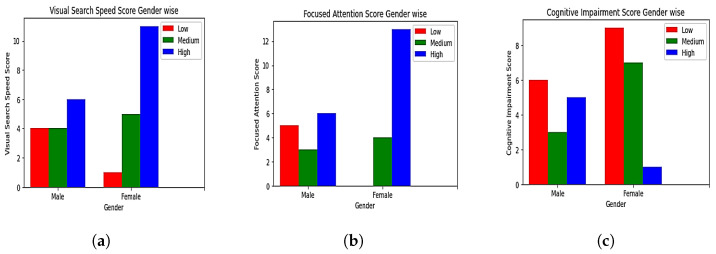
Gender-wise analysis of (**a**) visual-search-speed scores, (**b**) focused-attention scores, (**c**) overall cognitive-impairment scores.

**Figure 13 sensors-23-06848-f013:**
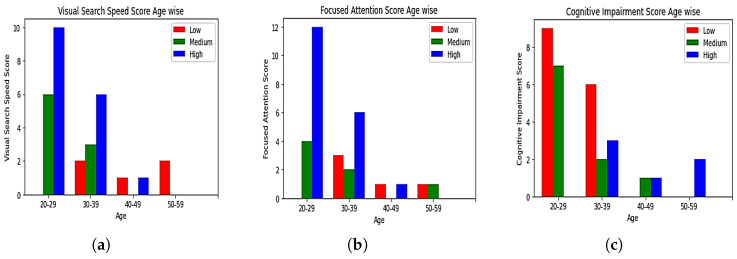
Age-wise analysis of (**a**) visual-search-speed scores, (**b**) focused-attention scores, (**c**) overall cognitive-impairment scores.

**Figure 14 sensors-23-06848-f014:**
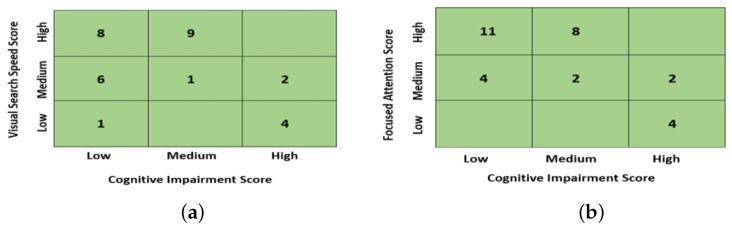
(**a**) Distribution of participants in cognitive-impairment and visual-search-speed scores. (**b**) Distribution of participants in cognitive-impairment and focused-attention scores. The numbers in each specific box represent the count of participants falling into the corresponding category.

**Figure 15 sensors-23-06848-f015:**
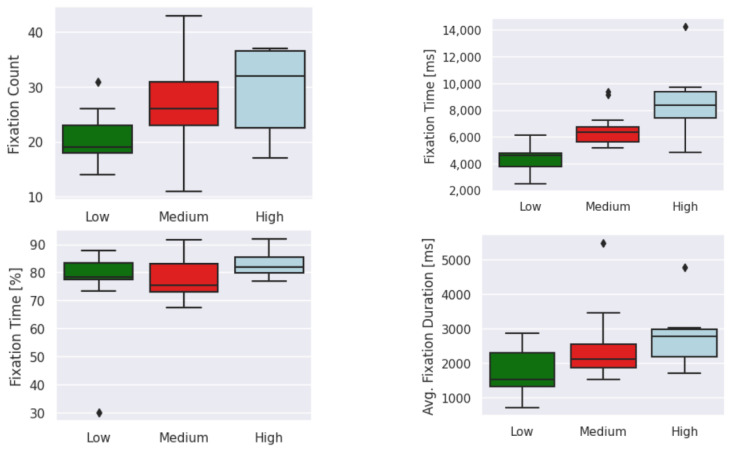
Boxplot of low-level features based on cognitive-impairment score. Outliers (⧫).

**Figure 16 sensors-23-06848-f016:**
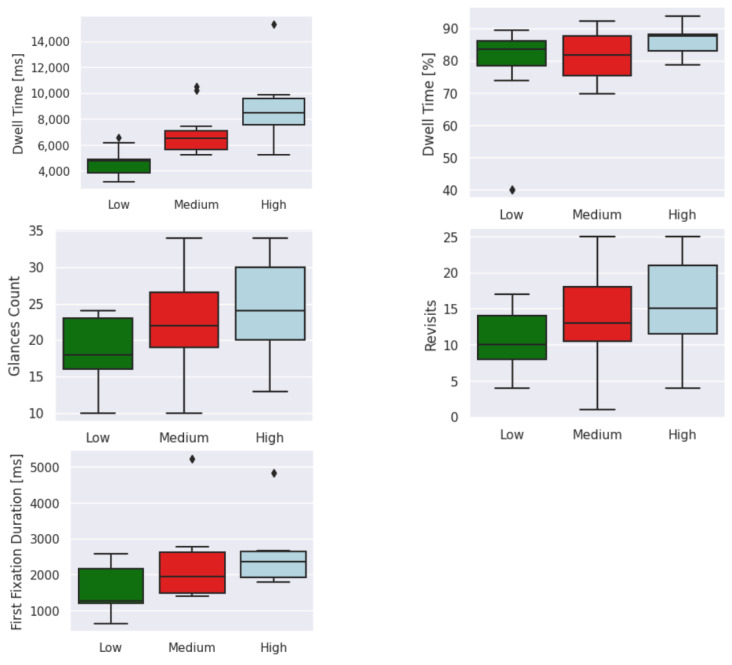
Boxplot of middle-level features based on cognitive-impairment score. Outliers (⧫).

**Figure 17 sensors-23-06848-f017:**
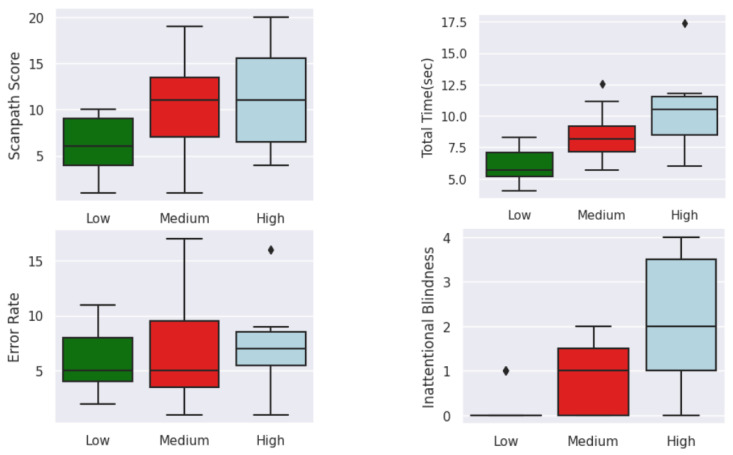
Boxplot of high-level features based on cognitive-impairment score. Outliers (⧫).

**Table 1 sensors-23-06848-t001:** Cognitive impairment and associated factor analysis.

Paper	Disease	Cognitive Impairment	Stimulus	Standard Cognitive Assessment Tools	Shortcomings of Standard Assessment Tools	Eye-Tracking Measures	Observations
[24]	ALS	Motor impairment	Visual paired- comparison task	ECAS	Cannot handle lower motor neuron atrophy	Antisaccadic error rate, saccadic latency	Higher antisaccades, error rate, saccadic latency
[7] [12] [24] [31] [32] [33]	AD	Memory impairment, impaired visual attention, attentional disengagement	Working memory tasks, deductive reasoning, memory recall task, visual memory task	ADAS-Cog, MMSE, MOCA	Longer time duration, not simple, subjects may feel highly stressed	Saccades, fixations, smooth pursuit	Longer time to fixate the target, shorter fixation duration, imprecise saccadic movements
[24] [57]	PD	Focused attention impairment, movement impairment, memory impairment	Saccadic task, TMT	SCOPA-COG PD-CRS, MOCA	Low sensitivity in recognizing cognitive deficits in the early stages of PD	Pupil diameter	Ocular abnormalities, longer response time
[40] [41] [43]	MCI	Memory impairment	Visual paired- comparison task, animal fluency, WLM, constructional praxis, TMT, Digit Span Subtest, Clock Drawing Test	MMSE, MOCA	Expensive, invasive, cannot detect early stages of the disease	Fixations, saccades, re-fixations, pupil diameter, total looking time, fixation count, percentage looking time on novel image	Percentage time in viewing the novel pictures could differentiate control group from MCI group
[24]	MS	Impairment of attention, executive function impairment, memory impairment	Saccadic task, ocular working memory task	MRI, MACFIMS	A trained evaluator needs at least 90 min for a full evaluation	Fixations, saccadic latency	Fixation instability, higher saccadic error rates, impaired pursuit
[24]	Epilepsy	Neuropsychological impairment	Vision-guided saccades, antisaccadic response inhibition, prosaccades, antisaccadic tasks	ET, PNS	Limited sensitivity, unsuitability for repeated assessment, sole focus on one aspect of cognition	Saccades, fixations	Increased error rate, longer reflexive time at the start of saccades

MACFIMS—Minimal Assessment of Cognitive Function in Multiple Sclerosis; ET—Epitrack; PNS—Portland Neurotoxicity Scale; WLM—Word List Memory.

**Table 2 sensors-23-06848-t002:** Eye-tracking features for the detection of cognitive impairment in different diseases.

Feature	Impairment	Disease
Inattentional blindness	Memory	PD, AD, MCI
Error rate	Memory, imprecise saccadic movement	AD, ALS, MCI, MS
Total completion time	Memory	AD, PD, MCI
Scanpath comparison score	Visual attention, diminished visual curiosity, abnormal visuospatial behavior	AD
Shorter fixation duration	Attentional disengagement	AD, MS
Higher saccadic latency	Motor impairment	ALS
Fixation instability	Impaired mobility and cognition	MS
Impaired pursuit	Impaired mobility and cognition	MS

PD—Parkinson’s disease; AD—Alzheimer’s disease; MCI—mild cognitive impairment; ALS—amyotrophic lateral sclerosis; MS—multiple sclerosis.

**Table 3 sensors-23-06848-t003:** Low-level, middle-level, and high-level features.

Type	Feature	Description
Low level	Fixation count (FL1)	Total number of fixation points
Low level	Fixation time (ms) (FL2)	Sum of the time duration on each fixation point
Low level	Fixation time (%) (FL3)	Percentage of fixation time with respect to total time
Low level	Fixation duration Average (ms) (FL4)	Average of all the fixation durations concerning the trial
Middle level	Dwell time (FM1)	Amount of time that respondents spend looking at a particular AOI
Middle level	Dwell time (%) (FM2)	Percentage of dwell time with respect to total time
Middle level	Glances count (FM3)	Count of fixation points in an AOI
Middle level	Revisits (FM4)	Number of times a participant returns their gaze to a particular spot or AOI
Middle level	First fixation duration ms (FM5)	Time duration of first fixation in an AOI
High level	Scanpath score (FH1)	Score based on the comparison of each participant’s scanpath and expected scanpath
High level	Total time (FH2)	Total completion time
High level	Error rate (FH3)	Rate of mistakes during the TMT task
High level	Inattentional blindness (FH4)	Indication of presence of inattentional blindness

**Table 4 sensors-23-06848-t004:** Parameters of visual-search-speed FIS and focused-attention FIS. (FIS-1 and FIS-2 submodules in ETMT model).

Parameter	Description
Fuzzy structure	Mamdani
Membership function	Trapezoidal
Number of membership functions for each input	3
Number of inputs	2
Number of outputs	1
Rules generated	9

**Table 5 sensors-23-06848-t005:** Parameters of ANFIS (submodule in ETMT model) for overall cognitive-impairment score.

Parameter	Description
Fuzzy structure	Sugeno
Membership function	Gaussian
Number of membership functions	3
Number of inputs	13
Number of outputs	1
Optimization method	Hybrid
Training number of epochs	30
Training samples	75%
Testing samples	25%

**Table 6 sensors-23-06848-t006:** Error statistics of ANFIS model (submodule in ETMT model) for cognitive-impairment score.

Error Statistic	Testing for Epoch Number = 20	Testing for Epoch Number = 30	Testing for Epoch Number = 40	Testing for Epoch Number= 50
RMSE	0.6702	0.3581	0.6504	0.8041
Error mean	−0.0851	0.1262	−0.0988	0.1835
Error STD	0.7107	0.3583	0.6873	0.8369

**Table 7 sensors-23-06848-t007:** Validation of clustering and labeling the data.

Machine Learning Algorithm	Testing Accuracy
Decision tree	100%
Linear discriminant	75%
Neural network	87.5%
KNN	87.5%
SVM	75%
Naive Bayes	87.5%

**Table 8 sensors-23-06848-t008:** Correlation of Eye-Tracking TMT with traditional TMT method.

Stimulus	Correlation Score	*p*-Value
TMT A Simple	0.727	0.003
TMT A Complex	0.769	0.001
TMT B Simple	0.734	0.002
TMT B Complex	0.725	0.003

**Table 9 sensors-23-06848-t009:** Cognitive impairment and associated standard tests.

	Motor Impairment	Visual Attention	Attentional Disengagement	Memory	Neuropsychological Impairment	Social Cognition	Executive Functioning	Processing Speed
ETMT	✓	✓	✓	✓	✓		✓	✓
ECAS	✓	✓		✓		✓		
MMSE		✓	✓	✓				
ADAS-Cog		✓	✓	✓	✓		✓	
SCOPA-COG	✓	✓	✓	✓	✓		✓	
PD-CRS	✓		✓	✓	✓			
MoCA		✓	✓	✓			✓	
MACFIMS		✓	✓	✓			✓	✓
MRI and CT scans	✓			✓	✓			

**Table 10 sensors-23-06848-t010:** Shortcomings associated with standard tests.

	Expensive	Need for Trained Evaluator	Difficult Detection During Early Stages of the Diseases	Longer Time Duration for the Tests	Cannot Handle Lower Motor Neuron Atrophy	Makes the Participant Highly Stressed
ETMT model						
Traditional TMT		✓				
ECAS		✓	✓	✓	✓	
MMSE			✓			
ADAS-Cog				✓		✓
SCOPA-COG			✓			
PD-CRS			✓			
MoCA			✓			
MACFIMS		✓		✓		
MRI and CT scans	✓	✓		✓		

**Table 11 sensors-23-06848-t011:** Time duration of ETMT and traditional TMT.

		ETMT	Traditional TMT
TMT-A simple	Min time (ms)	4042	3800
	Max time (ms)	17,402.8	10,200
TMT-A complex	Min time (ms)	16,821.8	12,000
	Max time (ms)	60,461.3	52,000
TMT-B simple	Min time (ms)	3654.9	3500
	Max time (ms)	17,202.6	19,900
TMT-B complex	Min time (ms)	33,438	18,200
	Max time (ms)	108,670.7	102,100
Entire test duration	Min time (ms)	66,619.1	37,500
	Max time (ms)	167,504.9	184,200

## Data Availability

Not applicable.

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
