# Peer review of "ETMT: A Tool for Eye-Tracking-Based Trail-Making Test to Detect Cognitive Impairment"

_sensors, 2023, doi:10.3390/s23156848_

Round 1

Reviewer 1 Report

The authors developed an iterested tool used for early detection of cognitive impariment. The results are promising. The paper presents these results in an extensive manner.  

In section Section 2.3 - Cognitive impaoirment assoc with diseases - please include some theoretical data regarding others diseases including, but not only : stroke, ocular diseass (paralysises, Age-related macular degeneration (AMD), etc)

Row 145 - please explain PD abreviation (probably Parkinson Disease), as it is used first time in the text

Row 197 - please explain AD abreviation, as it is used first time in the text

Row 206 - please explain TMT abreviation, as it is used first time in the text

Table 2- please explain the abreviations used in this table

Inclusion and exclusion criteria of study sample are not described. Please rectify it.

Figure 4 - please explain the abreviation used 

Row 209 - please explain FM5, FM1, as they are used first time in the text

Row 297 - please explain FL1 to FL4 - and please rectify this aspect for all abreviation used in the entire text!

In Section 3 - please include the cognitive test perfomed on the healthy subjects (concomitant with the described technology)

Results are extensive, with relevand graphics. Thet would benefit from some comparition between cognitive imparment score using the presented method and the cognitive score using a classical tool (already validated).

Minor editing of English language required

Reviewer 2 Report

Introduction:

In your introduction (especially in the first paragraph), you need to cite anything that is not common knowledge and not just extrapolate. 

When talking about the commonly used neuropsychological tests, please provide examples.

In your final paragraph please provide the objectives of this study instead of what you did.

Related works:

The Trail-making test part B can be quite simple and it is not time consuming. How are you defining time consuming? I can understand how MMSE, ADAS-Cog and FAB are time consuming, but the TMT-B takes < 1 minute if administered correctly. You can perhaps mention the MoCA in this section.

What does detecting personality traits have to do with detecting cognitive impairments?

I think later when you talk about the limitations of the TMT-B it is a good justification for why eye-tracking is needed, but definitely not time.

Methodology:

How were the participants recruited? What was the inclusion/exclusion criteria? What made you determine that 31 was the number needed?

You make several statements in your methods section that are not common knowledge. Please make sure that you cite them.

Results:

While the generations of these scores is great, perhaps the best way to use these scores is to identify where the deficits are instead of trying to generate an overall score?

Discussion:

I think this manuscript has value, but the messaging needs to be changed. My suggestion would be to use eye-tracking as a way to enhance the feedback practitioners can receive from the TMT-A/B by identifying aspects of cognition that may not be optimal. Otherwise the messaging in this manuscript is all over the place. Please have a clear message.

Round 2

Reviewer 2 Report

I think you've done a much better job in having a more coherent message.